# CLDN1 Sensitizes Triple-Negative Breast Cancer Cells to Chemotherapy

**DOI:** 10.3390/cancers14205026

**Published:** 2022-10-14

**Authors:** Marine Lemesle, Marine Geoffroy, Fabien Alpy, Catherine-Laure Tomasetto, Sandra Kuntz, Isabelle Grillier-Vuissoz

**Affiliations:** 1CRAN, UMR 7039, Université de Lorraine, 54506 Vandoeuvre-lès-Nancy, France; 2Institut de Génétique et de Biologie Moléculaire et Cellulaire (IGBMC), Institut National de la Santé et de la Recherche Médicale (INSERM), U1258, Centre National de la Recherche Scientifique (CNRS), UMR7104 and Université de Strasbourg, 67400 Illkirch, France

**Keywords:** breast cancer, TNBC, chemotherapy, CLDN1, sensitivity, apoptosis, biomarker

## Abstract

**Simple Summary:**

Triple-negative breast cancer (TNBC) treatment represents a major challenge in oncology. TNBC evolves into chemotherapy resistance for 60 to 70% of the patients. About 77% of the TNBC displays a lack of claudin-1 (CLDN1), a major tight junction component. We demonstrated that CLDN1 increased the sensitivity of TNBC cell lines to the main chemotherapeutic agents commonly used for breast cancer treatment. Our data support the idea that CLDN1 may be a good predictive chemotherapy response marker to help therapeutic management of TNBC patients. In longer terms, this study could allow new treatment protocols creation aimed to induce CLDN1 expression in TNBCs to increase their sensitivity to chemotherapy.

**Abstract:**

Triple-negative breast cancer (TNBC) is an aggressive subtype that constitutes 15–20% of breast cancer cases worldwide. Current therapies often evolve into chemoresistance and lead to treatment failure. About 77% of the TNBC lacks claudin-1 (CLDN1) expression, a major tight junction component, and this absence is correlated with poorer prognostic. Little is known about CLDN1 role on the chemosensitivity of breast cancer. Our clinical data analysis reveals that CLDN1 low expression is correlated to a poor prognostic in TNBC patients. Next, the sensitivity of various TNBC “claudin-1-high” or “claudin-1-low” cells to three compounds belonging to the main class of chemotherapeutic agents commonly used for the treatment of TNBC patients: 5-fluorouracil (5-FU), paclitaxel (PTX) and doxorubicin (DOX). Using RNA interference and stable overexpressing models, we demonstrated that CLDN1 expression increased the sensitivity of TNBC cell lines to these chemotherapeutic agents. Taken together, our data established the important role of CLDN1 in TNBC cells chemosensitivity and supported the hypothesis that CLDN1 could be a chemotherapy response predictive marker for TNBC patients. This study could allow new treatment protocols creation aimed to induce CLDN1 expression in TNBCs to increase their sensitivity to chemotherapy.

## 1. Introduction

Triple-negative breast cancer (TNBC) is an aggressive subtype that constitutes 15–20% of breast cancer cases worldwide [1]. TNBC is an heterogenous tumors group characterized by the lack of estrogen receptor (ER), progesterone receptor (PR) and HER2 receptor and therefore is not eligible for such targeted therapies. These tumors are associated with high recurrence rates and high metastasis incidence. Disease progression and recurrence typically occur within the first 3–5 years after diagnosis and distant metastases occurred much commonly to the brain and lung [2,3]. Current treatments for TNBC are mainly the combination of surgery, conventional chemotherapy and radiotherapy [4]. Nevertheless, the tumor evolves into chemotherapy resistance in about 60 to 70% of the patients, leading to a poor overall survival [1,5,6].

About 77% of the TNBC displays a lack of claudin-1 (CLDN1) expression, a major tight junction component [7,8]. CLDN1 plays a key role in cell adhesion maintenance, cell polarity and in paracellular permeability regulation [9,10,11]. CLDN1 is a transmembrane protein containing four membrane-spanning regions, two extracellular loops and N- and C-terminal cytoplasmic domains. The C-terminal domain is important for CLDN1 anchoring into tight junctions, it includes a PDZ domain binding motif which binds intracellular protein and is involved in signal transduction [9]. Most studies reported that a loss of CLDN1 expression correlates with increased malignant potential, invasiveness and high recurrence in invasive breast carcinoma patients, specifically in TNBC cases [12,13,14,15]. Although CLDN1 role in breast cancer is still ill-defined, several in vitro studies agree to the notion that CLDN1 is a tumor suppressor [16]. Indeed, CLDN1 downregulation was associated with a more malignant phenotype in non-invasive T-47D breast cancer cells [17]. Reciprocally, CLDN1 forced expression in TNBC cells decreased cell viability, inhibited cell migration and increased cell aggregation [18,19]. Moreover, CLDN1 expression induced apoptosis in MDA-MB-361 breast cancer cells and in MDA-MB-231 and Hs578T, two TNBC “claudin-1-low” cell lines [20,21]. There is a lack of consensus regarding the function of CLDN1 on chemosensitivity. In cancerous liver cells HepG2, CLDN1 silencing increased 5-FU sensitivity by inhibiting autophagy [22]. In A549 lung cancer cells, reduced CLDN1 expression also decreased cisplatin resistance [23]. In contrast, cisplatin sensitivity appeared to be correlated to CLDN1 expression in KLE endometrial carcinoma cells, by promoting the activation of the extrinsic apoptosis pathway [24]. CLDN1 was also described as a potential predictive marker for chemotherapy response for lung adenocarcinoma. In fact, CLDN1 expression induced by HDAC inhibitors in various low CLDN1-expressing lung adenocarcinoma cells increases their sensitivity to cisplatin [25]. Little is known on CLDN1 role on the chemosensitivity in breast cancer. Interestingly, Zhou et al., showed that CLDN1 overexpression increased sensitivity to etoposide, tamoxifen and cisplatin in the breast cancer hormone-dependent MCF-7 cell line [26]. In the present study, we aimed to identify the role of CLDN1 expression on “claudin-1-low” TNBC cancer cell response to chemotherapy. To address this question, we analyzed clinical data in Breast Cancer Gene-Expression database. The study released that CLDN1 low expression is correlated to a poor prognostic in TNBC patients. Next, the sensitivity of various TNBC “claudin-1-high” or “claudin-1-low” cells to three compounds belonging to the main class of chemotherapeutic agents commonly used for the treatment of TNBC patients: 5-fluorouracil (5-FU), paclitaxel (PTX) and doxorubicin (DOX). Using RNA interference and stable overexpressing models, we demonstrated that CLDN1 expression increased the sensitivity of TNBC cell lines to these chemotherapeutic agents. 

Taken together, the results established the important role of CLDN1 in TNBC breast cancer cells chemosensitivity and contribute to better understand its mechanism of action. Our data support the hypothesis that CLDN1 could be a chemotherapy response predictive marker for TNBC patients.

## 2. Materials and Methods

### 2.1. Cell Culture and Treatment

The human TNBC cell lines were cultured at 37 °C under 5% CO_2_. MDA-MB-231 and HCC1806 cells were cultured in phenol red RPMI (Sigma-Aldrich, St. Quentin Fallavier, France), supplemented with 10% FBS (Sigma-Aldrich) and 2 mM L-glutamine (Sigma-Aldrich). Hs578T cells were grown in DMEM medium containing phenol red (Sigma-Aldrich) supplemented with 10% FBS (Sigma-Aldrich), 2 mM L-glutamine (Sigma-Aldrich) and 10 µg/mL bovine insulin (Sigma-Aldrich). All cell lines were purchased from the ATCC (American Type Culture Collection). Cells were treated for 72 h with DMSO or Ethanol (vehicle) or different concentrations of 5-FU (F6627, Sigma-Aldrich), PTX (T7402, Sigma-Aldrich) and DOX (D1515, Sigma-Aldrich). All chemotherapeutic agents were dissolved in DMSO (5-FU), ethanol 100% (PTX) or sterile water (DOX) and were stored at −20 °C.

### 2.2. Generation of Stable Cell Lines

MDA-MB-231 overexpressing CLDN1 called MDA-MB-231/CLDN1 and control cells stably transfected by empty plasmid MDA-MB-231/CTRL were constructed in the laboratory [19]. The Hs578T/CLDN1 and Hs578T/CTRL cell lines were established by retroviral infection. The Hs578T cell line was transfected with the retroviral vectors pQCXIP as control or pQCXIP-CLDN1. Cells were subsequently selected for 2 weeks by exposure to puromycin at 50 ng/mL (ThermoFisher Scientific, Illkirch-Graffenstaden, France).

### 2.3. RNA Interference Transfection

MDA-MB-231 (2 × 10^5^ cells/well) and HCC1806 (1 × 10^5^ cells/well) cells were plated in 6-well culture plates for 24 h. For siRNA transient transfection, OligofectamineTM Reagent (ThermoFisher Scientific, Illkirch-Graffenstaden, France) were used according to the manufacturer’s protocol. siRNAs sequences against human CLDN1 are 5′-GCAAUAGAAUCGUUCAAGA-3′, 5′-GGCAUGAAGUGUAUGAAGU-3′, 5′AGAAUUCUAUGACCUAUG-3′ and 5′-ACGAUGAGGUGCAGAAGAU-3′. The siRNA concentration was 2 nM for MDA-MB-231 cells (MDA-MB-231 scRNA or MDA-MB-231 siCLDN1) and 12 nM for the HCC1806 cells (HCC1806 scRNA or HCC1806 siCLDN1). After 4 h, siRNA transfection was stopped by adding 10% FBS and cells were treated by chemotherapeutics agents 7 h after transfection for 72 h. 

### 2.4. Western Blot

MDA-MB-231 (1.6 × 10^5^ cells/well), Hs578T and HCC1806 (0.8 × 10^5^ cells/ well) cells were seeded in 6-well plates. After 72 h of treatment, cultured media were collected as well as scraped cells at 4 °C. Tubes were centrifuged at 145× *g* at 4 °C for 10 min. Supernatants were removed and lysis buffer (0.1× Triton, 0.25% sodium deoxycholate, 50 µM Tris pH 7.4, 150 mM NaCl, 1 mM EDTA, phosphatase and protease inhibitors cocktail (Sigma-Aldrich)) was added to solubilize pellet for 30 min at 4 °C and then centrifuged at 20,124× *g* for 5 min at 4 °C. Protein concentration was determined using the Bradford method (Bio-Rad Protein Assay, BioRad, Marnes-la-Coquette, France). Next, 50 µg of protein were denatured in Laemmli buffer (125 mM Tris–HCl, pH 6.8, 10% SDS, 50% saccharose, 0.5 M DTT, 0.05% bromophenol blue) by heated at 95 °C for 5 min. For each sample, total protein samples were separated by SDS–PAGE. After electrophoresis, proteins were transferred onto nitrocellulose membrane. Nonspecific binding sites were blocked in TNT buffer (50 mM Tris–HCl, 150 mM NaCl, 0.1% Tween 20) with 5% nonfat powder milk and membranes were then incubated with the primary antibodies diluted in blocking solution overnight at 4 °C. The anti-cleaved caspase 7 antibody (#9494S, Cell Signaling Technology, Saint-Cyr-L’École, France) was diluted at 1:800, the anti-cleaved PARP-1 antibodies (552596, BD Biosciences, Le Pont de Claix, France) and anti-CLDN1 (C5838-02C, US Biological, Nanterre, France) were diluted at 1:1000 and the GAPDH antibody (60004-1-IgG, ProteinTech, Illkirch-Graffenstaden, France) was diluted at 1:5000. Next, membranes were probed with secondary antibodies anti-Mouse or anti-Rabbit IgG–Peroxidase antibody produced in goat (A4416 or A9169, Sigma Aldrich) for 1 h at room temperature. The immunostaining was observed using a Chemidoc Touch Imaging System (BioRad) and quantification of band intensity was performed using Image Lab Analysis Software (BioRad). Ratio between the intensity of the band of interest and GAPDH was calculated to determine the protein level of induction after treatment.

### 2.5. Cell Viability

MDA-MB-231 (0.32 × 10^5^ cells/well), Hs578T and HCC1806 (0.16 × 10^5^ cells/well) were seeded in 24-well plates. Cell viability was determined using ATP assay according to the manufacturer’s protocol (CellTiter-Glo TM Luminescent Cell Viability Assay, Promega, Charbonnières-les-Bains, France). Cell viability percentage were calculated by reporting control cell viabilities to 100%.

### 2.6. Flow Cytometry

MDA-MB-231 (1.6 × 10^5^ cells/well), Hs578T and HCC1806 (0.8 × 10^5^ cells/ well) cells were seeded in 6-well plates. After 72 h of treatment, cells were labeled by FITC-conjugated Annexin V (556419, BD Pharmingen, Illkirch-Graffenstaden, France) and propidium iodide (PI) (P3566, Life Technologies, Illkirch-Graffenstaden, France). Cell staining was performed according to the manufacturer’s protocol. Cells were then analyzed on CytoFLEX cytometer (Beckman, Villepinte, France) and the results were analyzed by using cytoFLEX software (Beckman). The percentage of early-apoptotic cells corresponds to cells labeled with annexin V, cells in late apoptosis or dead cells are co-labeled with annexin V and PI.

### 2.7. CLDN1 mRNA Expression Raincloud Analysis in Breast Tumors

CLDN1 mRNA level expression in breast cancers was assessed with the website Breast Cancer Gene-Expression Miner v4.8 (bcGenExMiner v4.8) (http://bcgenex.ico.unicancer.fr/) (accessed on 29 September 2022) [27]. For this purpose, CLDN1 mRNA expression raincloud plot for 2536 breast tumors was generated. The following parameters were selected: Exhaustive expression analysis; DNA microarrays Affymetrix^®^; CLDN1 highest probe (222549_at); Raincloud type of plot; TNBC (IHC) status. 

### 2.8. Kaplan–Meier Overall Survival Analysis

To assess the prognostic value of CLDN1 gene in TNBC patients, the Kaplan–Meier Plotter website (Budapest, Hungary) (https://kmplot.com/analysis/) (accessed on 29 September 2022) was used [28,29]. For this purpose, CLDN1 Kaplan–Meier plots for ERα-negative, PR-negative and HER2-negative breast cancer patients were generated. The following parameters were selected: mRNA (RNA-seq), overall survival (OS); split patients by upper quartile, follow up threshold: 240 months, we restricted analysis to subtypes ER negative, PR negative and HER2 negative patients and no endocrine treated patients. Different cohorts of patients were chosen for the analyzes: chemotherapy versus no chemotherapy treatment. Logrank *p*-values < 0.05 for the Kaplan–Meier (KM) plots of CLDN1 gene was considered statistically significant.

### 2.9. Statistical Analysis 

All data correspond to the mean ± standard error of the mean (SEM) of 3 to 8 independent experiments. Bars represent mean ± SEM. The statistical tests are performed with the Prism software (GraphPad). For the same cell line, the comparison between 2 conditions was determined by using the Student t test. Multiple comparisons were performed with ANOVA test followed by Post-hoc Dunnett test to compare the mean of the control condition with the means of all other conditions or Tukey’s Post-hoc test for all other comparisons. A *p*-value of less than 0.05 was considered statistically significant with * *p* < 0.05, ** *p* < 0.01, *** *p* < 0.001, **** *p* < 0.0001.

## 3. Results

### 3.1. CLDN1 Expression Is Associated with Better Sensitivity to Conventional Chemotherapeutic Agents

No data are available on the effect of CLDN1 expression level on TNBC cells chemosensitivity. Hence, we started by a bioinformatic study on clinical databases. First, we examined the distribution of CLDN1 mRNA expression in TNBC tumor versus non-TNBC tumor. The raincloud plot showed that CLDN1 mRNA expression was the most variable in the TNBC group compared to the non-TNBC group. Indeed, the TNBC group is composed of two distinct subpopulations: tumors expressing very high level of CLDN1 mRNA (“claudin-1-high”) and tumors expressing low level of CLDN1 mRNA (“claudin-1-low”) (Figure 1A). This finding led us to investigate whether this difference in CLDN1 expression in the TNBC group could induce a difference in the overall survival (OS) (Figure 1B). We performed OS Kaplan-Meier curve based on CLDN1 expression and presence and/ or absence of chemotherapy treatment. The analysis showed that patients with “claudin-1-high” tumors had significantly higher OS compared to “claudin-1-low” tumors in any condition (Figure 1B–D). Indeed, 80 months after diagnosis, only 50% of TNBC “claudin-1-low” patients survived against more than 80% for TNBC “claudin-1-high” patients (Figure 1B). Despite a similar trend of the OS in presence (Figure 1C) or in absence of chemotherapy (Figure 1D), it should be noted that the OS of “claudin-1-low” patients dropped less rapidly when they received chemotherapy. Thus, 50 months after diagnosis, OS of “claudin-1-low” patients remained at 80% against less than 40% in absence of chemotherapy, thereby suggesting a partial response to chemotherapy. More importantly, these data highlighted a strong correlation between the level of CLDN1 expression and TNBC patients OS.

Therefore, to verify whether there is a link between CLDN1 expression in TNBCs and chemosensitivity, we started our experimental study by analyzing the viability of a series of TNBC cells exposed to 5-FU, PTX and DOX. HCC1806 “claudin-1-high” cells and MDA-MB-231 and Hs578T “claudin-1-low” cells were exposed to increasing doses of 5-FU, PTX and DOX or vehicle (control cells) for 72 h, then the cell viabilities were measured by the quantification of the ATP level, which signals the presence of metabolically active cells. We determined the IC_50_ which corresponds to the concentration leading to a 50% decrease of the viable cell percentage. HCC1806 cells significantly displayed a higher 5-FU sensitivity (IC_50_ = 15.3 µM) compared to MDA-MB-231 (IC_50_ = 186.5 µM) and Hs578T (IC_50_ = 181.2 µM) (Figure 2A). Similar trend was observed with PTX, the sensitivity of HCC1806 (IC_50_ = 3.7 nM) was higher than MDA-MB-231 (IC_50_ = 18 nM) and Hs578T (IC_50_ = 5 nM) (Figure 2B). Once more, HCC1806 showed a significant better sensitivity to DOX treatment (IC_50_ = 27.5 nM) compared to MDA-MB-231 (IC_50_ = 305 nM) and Hs578T (IC_50_ = 210 nM) (Figure 2C). Of note, no data were obtained for higher concentration than 5 nM of PTX and 30 nM of DOX for HCC1806, as no cells survived. For each chemotherapeutic agent, the TNBC “claudin-1-high” HCC1806 cells were more sensitive than TNBC “claudin-1-low” MDA-MB-231 and Hs578T cells.

Since the three chemotherapeutic agents induced a decrease of cell viability in the three cell lines, we studied apoptosis by analyzing the activation of the executioner caspase 7 (cCASP7) and the cleavage of PARP-1 (cPARP) (a well-known caspase substrate) by western blot (Figure 2D–F). Consistent with the viability assay, activation of apoptosis appeared at lower dose for the HCC1806 cell line (25 µM for 5-FU, 1 nM for PTX and 25 nM for DOX) than in MDA-MB-231 cells (150 µM 5-FU, 2.5 nM PTX and 150 nM DOX) and in Hs578T cells (100 µM for 5-FU, 2.5 nM for PTX and 50 nM for DOX) (Figure 2D–F). All together these data showed that “claudin-1-high” HCC1806 cells were more sensitive to apoptosis than the “claudin-1-low” MDA-MB-231 and Hs578T cells to the three chemotherapeutic agents. Next, to confirm this point, we characterized apoptosis by annexin V/PI co-staining. To compare the sensitivity of the three cell lines, a common dose of chemotherapeutic compound was chosen: 50 µM 5-FU, 5 nM PTX and 86 nM DOX. The common concentrations were used as an optimal killing dose for the sensitive HCC1806 cells (20% of viable cells) (Figure 2), allowing analysis of the proapoptic effect in both “claudin-1-high and -low” models. Following 72 h of treatment, early-apoptotic and late-apoptotic/dead cells were quantified by flow cytometry (Figure 3 and Appendix A). Staurosporine was used as positive apoptosis control. 5-FU led to a significant increase of HCC1806 late-apoptotic cell percentage (24.12%). In contrast, 5-FU did not induce any significant apoptosis in MDA-MB-231 and Hs578T cells (Figure 3A). Similar to 5-FU, PTX induced a potent rise of the HCC1806 cells in late apoptosis reaching 59.63% of the cells and it slightly enhanced the percentage of early-apoptotic cells (8%). In MDA-MB-231 and Hs578T cells, PTX significantly increased the percentage of early-apoptotic cells to reach 28.16% and 37.39%, respectively. In Hs578T cells, we also observed an induction of the late apoptosis in response to PTX (Figure 3B). Once again, in response to DOX, the most significant effect was obtained in HCC1806 cells with 23.05% of late-apoptotic cells *versus* a non-significant induction in MDA-MB-231 cells and a 19.84% of late-apoptotic Hs578T cells (Figure 3C). All these results confirmed that chemotherapeutic agent stimulated apoptosis more efficiently in the HCC1806 “claudin-1-high” cells compared to the “claudin-1-low” MDA-MB-231 and Hs578T cells. 

As CLDN1 expression was sufficient to induce apoptosis in several “claudin-1-low” cell lines [20,21], we then hypothesized that the elevated level expression of CLDN1 in HCC1806 cells could contribute to their higher sensitivity to chemotherapeutic compounds. Therefore, we examined CLDN1 level expression together with apoptotic response in treated cells by western blot (Figure 2D–F). In response to treatment, CLDN1 expression remained elevated in HCC1806 cells and not detectable in Hs578T cells. Of note, in MDA-MB-231 cells, PTX and DOX induced CLDN1 expression while 5-FU had no effect highlighting a novel correlation between CLDN1 expression and apoptosis induction in MDA-MB-231 cells (Figure 2D–F).

In summary, these data showed that HCC1806 cells, which presents a high level of CLDN1, were more sensitive to chemotherapeutic agents compared to both “claudin-1-low” MDA-MB-231 and Hs578T cell lines.

### 3.2. CLDN1 Is Involved in the Chemosensitivity of HCC1806 “Claudin-1-High” Cells to 5-FU, PTX and DOX

Remarkably, HCC1806 cells are sensitive to the three chemotherapeutic agents although they display different mechanisms of action. 5-FU and DOX acts on DNA whereas PTX acts on cytoskeletal proteins. 5-FU is an antimetabolic drug which interferes with nucleoside metabolism and can be incorporated into DNA and RNA leading to cytotoxicity [30]. DOX is a DNA intercalator which forms complexes with DNA and inhibits topoisomerase II activity [31]. PTX is a mitotic inhibitor which promotes the assembly of tubulin into microtubules and prevents their dissociation leading to mitotic arrest [32]. All these chemotherapeutic compounds induced apoptosis of HCC1806 “claudin-1-high” cells, so we raised the hypothesis that CLDN1 could be involved in their chemosensitivity.

To determine whether CLDN1 was responsible for the higher chemosensitivity of HCC1806 cells, we inhibited its expression by RNA interference (siCLDN1) (Figure 4A). Next, HCC1806 cells were treated with an optimal killing dose of each compound. The concentrations were chosen to decreased cell viability up to 80%: 50 µM of 5-FU (Figure 4B–D), 5 nM of PTX (Figure 4E–G) and 86 nM of DOX (Figure 4H–J) were used for 72 h treatment. By western blotting, we confirmed that CLDN1 expression in HCC1806 siCLDN1 cells was strongly decreased compared to the control cells (scRNA) (Figure 4A) and under chemotherapeutic treatment (Appendix A). For each agent, we first quantified cell viability by measuring the ATP level; then, we analyzed apoptosis by flow cytometry (Appendix A) and western blot. In response to 5-FU, we observed 8.42% and 22.72% of viable HCC1806 cells in control and CLDN1 silencing conditions, respectively (Figure 4B). Consistent results were obtained by flow cytometry, CLDN1 silencing significantly increases the percentage of living cells compared to HCC1806 scRNA cells under 5-FU treatment (Figure 4C). CLDN1 expression inhibition in HCC1806 siCLDN1 cells also led to a significant decrease of the late-apoptotic cell percentage from 39.86% in HCC1806 scRNA cells to 26.11% in HCC1806 siCLDN1 cells (Figure 4C). However, we were unable to detect significant difference level of cleaved of PARP-1 and CASP7 by western blotting (Figure 4D). These data showed that CLDN1 is involved in the sensitivity of HCC1806 cells to 5-FU.

Considering PTX, the cell viability rose when CLDN1 expression was inhibited (Figure 4E) and the percentage of living cells was increased from 27.06% in HCC1806 scRNA cells to 37.44% in HCC1806 siCLDN1 (Figure 4F). The inhibition of CLDN1 expression led also to significant decrease of the percentage of early- (5.18% to 2.73%) and late-apoptotic cells (61.21% to 51.53%) (Figure 4F). This result was confirmed by western blotting: inhibition of CLDN1 expression was associated to a 3.23-fold and a 5-fold decrease of cleaved PARP-1 and CASP7, respectively (Figure 4G). These data also highlighted that CLDN1 is involved in HCC1806 cells PTX sensitivity.

In the presence of DOX, inhibition of CLDN1 expression increased cell viability from 17.04% in HCC1806 scRNA cells to 32.26% in HCC1806 siCLDN1 (Figure 4H) and led to a significant rise of the living cells percentage (51.97% to 77.95%) (Figure 4I). Inhibition of CLDN1 expression also induced a drop of late apoptosis (34.40% to 14.89%) (Figure 4I). Under this condition, significant decrease of cPARP (4.58-fold) and cCASP7 (4.1-fold) were observed (Figure 4J). Again, reduction of CLDN1 expression decreased the sensitivity of the HCC1806 cells to DOX. 

Taken together, the reduction of CLDN1 expression in “claudin-1-high” HCC1806 cells increased cell viability and decreased apoptotic effect of 5-FU, PTX and DOX treatments. All these results showed that CLDN1 is involved in the sensitivity to the three chemotherapeutic agents in HCC1806 cells. Since silencing of CLDN1 makes “claudin-1-high” HCC1806 cells more resistant to chemotherapeutic agents, we decided to study if endogenous or ectopic CLDN1 expression could enhance the chemosensitivity of “claudin-1-low” MDA-MB-231 and Hs578T TNBC cells.

### 3.3. Endogenous CLDN1 Sensitizes MDA-MB-231 Cells to PTX and DOX

In “claudin-1-low” cells, we observed that chemotherapy had a distinct impact on the endogenous CLDN1 expression level: while none of the chemotherapeutic agents induced CLDN1 expression in Hs578T cells, DOX and PTX did it in MDA-MB-231 cells. To study whether endogenous CLDN1 expression modulates the sensitivity to PTX and DOX, we silenced its expression in MDA-MB-231 cells and measured apoptosis (early and late) (Appendix A) in silenced MDA-MB-231 siCLDN1 and control MDA-MB-231 scRNA cells in response to optimal concentrations previously determined of both agents (5 nM of PTX and 86 nM of DOX).

In MDA-MB-231 cells, RNA interference (MDA-MB-231 siCLDN1) decreased by 3.7-fold CLDN1 expression induced by PTX (Figure 5A and Appendix A). In the presence of PTX, cell viability was increased in MDA-MB-231 siCLDN1 cells compared to MDA-MB-231 scRNA cells (from 39.89% to 63.35%) (Figure 5B). MDA-MB-231 siCLDN1 cells displayed lower percentages of early- (7.74% versus 13.11%) and late-apoptotic cells (25.20% *versus* 42.14%) compared to MDA-MB-231 scRNA cells. In addition, a decrease (3.82-fold) of the cCASP7 in MDA-MB-231 siCLDN1 cells treated with PTX was observed. Nevertheless, no significant difference was observed for cPARP (Figure 5C). These data showed that CLDN1 expression is involved in the sensitivity of MDA-MB-231 to PTX.

In presence of DOX, CLDN1 expression was reduced by RNA interference by 5.54-fold in MDA-MB-231 siCLDN1 compared to MDA-MB-231 scRNA cells (Figure 5D and Appendix A). As for PTX, we observed an increase of cell viability between MDA-MB-231 scRNA and siCLDN1 cells treated with DOX (from 72.09% to 82.09%) (Figure 5E). This effect was accompanied by a decrease of early- and late-apoptosis compared to MDA-MB-231 scRNA (Figure 5E). Similarly, the cleavage of cPARP and cCASP7 dropped (by 2.44- and 1.72-fold, respectively) in MDA-MB-231 siCLDN1 cells compared to MDA-MB-231 scRNA cells (Figure 5F). CLDN1 expression is therefore implicated in the DOX sensitivity of MDA-MB-231 cells.

All these results demonstrated that endogenous CLDN1 expression is implicated in the PTX and DOX sensitivity of MDA-MB-231 cells.

### 3.4. Ectopic CLDN1 Expression Sensitizes Hs578T Cells to PTX and DOX but Not to 5-FU

To further demonstrate that CLDN1 favors chemotherapeutic response, we forced its expression in Hs578T cells. We generated Hs578T CLDN1 overexpressing cells (Hs578T/CLDN1) and measured apoptosis induction (Appendix A) and cell viability after treatment with IC_50_ concentrations of PTX (5 nM), DOX (210 nM) and 5-FU (181 µM) (Figure 2). As expected, Hs578T/CLDN1 cells displayed a strong increase of CLDN1 expression compared to the empty vector Hs578T/CTRL in control and treated cells (Figure 6A and Appendix A).

We first studied the effect of CLDN1 overexpression on PTX sensitivity. No significant effect on cell viability was observed in Hs578T/CLDN1 compared to Hs578T/CTRL for non-treated and PTX conditions (Figure 6B). However, CLDN1 expression was associated with an increase in the percentage of cells in early apoptosis in the presence of PTX (from 38.61% to 47.80%) (Figure 6B). In addition, more cleavage of cPARP and cCASP7 (4.1- and 1.87-fold, respectively) was observed in response to PTX in Hs578T/CLDN1 cells compared to Hs578T/CTRL cells (Figure 6C), supporting the idea that CLDN1 sensitizes Hs578T cells to PTX.

CLDN1 overexpression compromised cell viability under DOX treatment. Less living cells were measured in Hs578T/CLDN1 compared to Hs578T/CTRL cells (from 76.40% to 54.46%) (Figure 6D). Accordingly, in response to DOX, the percentage of early- and late-apoptotic cells was increased in the presence of CLDN1 (Figure 6D). Furthermore, the cleaved form of the apoptotic markers PARP-1 and CASP7 was enhanced (3.54- and 3.45-fold, respectively) in Hs578T/CLDN1 cells compared to Hs578T/CTRL cells (Figure 6E). These data showed that CLDN1 sensitizes Hs578T cells to DOX. 

In contrast, CLDN1 overexpression in Hs578T cells had no significant effect on 5-FU sensitivity. No difference in viability and early or late-apoptotic cells was observed between Hs578T/CLDN1 and Hs578T/CTRL treated or not to 5-FU (Figure 6F). In the same way, we detected no significant difference between apoptotic markers in western blot (Figure 6G). This data suggested that CLDN1 does not sensitize “claudin-1-low” Hs578T cells to 5-FU suggesting that 5-FU elicit a different biological response in these cells.

All together these data, showed that ectopic expression of CLDN1 sensitized Hs578T to PTX and DOX but not to 5-FU.

### 3.5. Ectopic CLDN1 Expression Sensitizes MDA-MB-231 Cells to 5-FU t

Next, we studied the implication of CLDN1 in the sensitivity to 5-FU in “claudin-1-low” MDA-MB-231 cells. In contrast to PTX and DOX treatments, 5-FU did not induce endogenous CLDN1 expression in MDA-MB-231 cells (Figure 2D). CLDN1 expression was forced in those cells, and we studied cell viability and apoptosis induction by flow cytometry (Appendix A) and western blot in response to 5-FU. We used MDA-MB-231 cells stably overexpressing CLDN1 (MDA-MB-231/CLDN1) previously developed in our laboratory [19]. Next, we treated MDA-MB-231 cells with 5-FU (50 µM), a suboptimal concentration which did not reduce viability and did not induce apoptosis (Figure 2A,D). MDA-MB-231/CLDN1 cells displayed a strong increase of CLDN1 expression compared to the empty vector MDA-MB-231/CTRL in control and 5-FU treated cells (Figure 7A and Appendix A).

As expected, CLDN1 overexpression alone significantly reduced cell viability (from 94.14% to 76.51%), increased early (from 0.70% to 2.63%) and late (from 3.31% to 14.10%) apoptosis in non-treated cells (Figure 7B). 5-FU also diminished the cells viability (from 81.66% to 61.52%) and increased percentages of early- and late-apoptotic cells reaching 8.22% and 24.15%, respectively (Figure 7B). Of note, in MDA-MB-231/CTRL cells, 5-FU had no impact on cell viability and on early- and late-apoptosis (Figure 7B). Interestingly, in MDA-MB-231/CLDN1 cells, the cell viability was significantly decreased (1.24-fold) and late apoptosis was more strongly induced in 5-FU treated cells than in non-treated cells (1.71-fold) (Figure 7B). These results were confirmed by western blot analysis. CLDN1 overexpression in non-treated cells induced a 12- and 10-fold increase of cPARP and cCASP7, respectively (Figure 7C). In 5-FU treated cells, the effect was accentuated with cPARP and cCASP7 increase of 14.2- and 7.2-fold, respectively. 5-FU apoptotic effect was always significantly more potent in MDA-MB-231/CLDN1 treated cells: the level of cPARP was higher in MDA-MB-231/CLDN1 5-FU treated cells than in MDA-MB-231/CLDN1 non-treated cells with a significant induction of 3.2-fold (Figure 7C). In fine, these results indicated that CLDN1 overexpression in MDA-MB-231 cells improved their sensitivity to 5-FU. 

## 4. Discussion

The TNBC subtype constitutes the most aggressive form of breast cancer and displays a poorer prognosis compared to other breast cancer subtypes. Triple negative tumors represent around 15 to 20% of breast cancer and displays poor prognosis. The main challenge in TNBC treatment is the lack of effective targeted therapy and the rapid appearance of chemoresistance. CLDN1, a major component of tight junctions, is under expressed in 77% of TNBC [7]. Low CLDN1 expression is correlated to the drastic drop of live expectancy [12]. Little is known on the role of CLDN1 on chemosensitivity. A previous study from our laboratory showed that CLDN1 overexpression induced apoptosis in TNBC “claudin-1-low” cells and potentialized the pro-apoptotic effect of an anticancer drug, Δ2-TGZ [20]. In this context, we analyzed if CLDN1 may improve the sensitivity to chemotherapies in TNBC cells. We studied the response to 5-FU, PTX and DOX which are the three commonly used chemotherapeutics agents used to treat these tumors. This is the first investigation on the effect of CLDN1 expression on TNBC chemosensitivity. 

Clinical data showed two distinct subpopulations in TNBC: “claudin-1-low” and “claudin-1-high” tumors and revealed a strong correlation between the level of CLDN1 expression and TNBC patients OS. The experimental study showed that the three chemotherapeutic agents were more effective to decrease cell viability in TNBC “claudin-1-high” HCC1806 cells compared to “claudin-1-low” MDA-MB-231 and Hs578T cells. In agreement with the viability results, apoptosis was more strongly induced in “claudin-1-high” cells compared to “claudin-1-low” cells in response to the three chemotherapeutic agents after 72 h of treatment. In HCC1806 cells, all chemotherapeutic agents induced the cleavage of PARP-1 and CASP7 at lower doses compared to MDA-MB-231 and Hs578T cells. This higher sensitivity of HCC1806 cells was also confirmed by Annexin V labeling and quantification by flow cytometry using comparison doses 5-FU, PTX and DOX. These data agree with the involvement of CLDN1 in chemosensitivity of TNBC cells.

Several studies showed that CLDN1 overexpression led to apoptosis in MDA-MB-361, MDA-MB-231 and Hs578T breast cancer cells [20,21]. We established that HCC1806 cells maintained a high level of CLDN1 expression regardless of chemotherapeutic treatment and maintained their strong chemosensitivity. Supporting the notion that CLDN1 expression favors sensitivity to chemotherapy, PTX and DOX induced endogenous CLDN1 expression in MDA-MB-231 “claudin-1-low” cells and drove to a higher apoptotic response than 5-FU which induce no CLDN1 expression or apoptosis. According to our data, low CLDN1 expression was previously associated with PTX-resistance in esophageal cancer cells [33]. Conversely, it was shown that CLDN1 expression increased the sensitivity of breast and lung cancer cells to various anticancer agents such as cisplatin, etoposide and tamoxifen [24,25,26].

However, the question of whether CLDN1 is clearly involved in the sensitivity of TNBC cells to chemotherapeutic agents remained unelucidated. Thus, to assess this question, we inhibited its expression by RNA interference in HCC1806 “claudin-1-high” cells and study their sensitivity to the three chemotherapeutic agents. Our results showed that downregulation of CLDN1 expression led to a significant drop of apoptosis in response to 5-FU, PTX and DOX and consequently increased resistance of HCC1806 cells to the three chemotherapeutic agents. These results reinforce the hypothesis that CLDN1 expression improves chemotherapeutic sensitivity. 

Next, we investigated if CLDN1 expression could sensitize TNBC “claudin-1-low” MDA-MB-231 and Hs578T cells to chemotherapeutic agents. Given that endogenous CLDN1 expression is induced by PTX and DOX in TNBC “claudin-1-low” MDA-MB-231 cells, we used RNA interference strategy. We demonstrated that endogenous CLDN1 expression increases MDA-MB-231 cells sensitivity to PTX and DOX.

Next, we studied if ectopic CLDN1 expression could enhance the sensitivity of TNBC “claudin-1-low” Hs578T cells to chemotherapeutic compounds. CLDN1 expression improved Hs578T sensitivity to PTX and DOX whereas it had no impact on the response to 5-FU. Since in MDA-MB-231 cells, 5-FU also does not induce CLDN1 expression, we stably expressed CLDN1 to study its contribution to the sensitivity to this chemotherapeutic agent. Remarkably, under ectopic CLDN1 overexpression 5-FU sensitivity was improved in MDA-MB-231 cells. All these results suggest a common mechanism of action between Hs578T and MDA-MB-231 cells regarding CLDN1 function on PTX and DOX response in contrast to 5-FU. We also previously reported that an anti-cancer drug (Δ2-TGZ) induced apoptosis differently in both cell lines: CLDN1 is involved in Δ2-TGZ apoptotic effect in MDA-MB-231 cells but not in Hs578T cells [20]. This difference of response to 5-FU is not related to CLDN1 level and may be explained by a lower 5-FU uptake in Hs578T cells. Indeed, the transporter-mediated uptake of 5-FU is negligible in this cell line compared to other TNBC cell lines including MDA-MB-231 cells [34]. As a perspective, this study could be extended using mouse tumors xenograft. For this purpose, the chemotherapeutic response would be monitored in CLDN1 high and CLDN1 low tumors.

How CLDN1 sensitizes these cells to chemotherapy remains unclear. To date CLDN1 had not been directly involved in apoptosis. Given that CLDN1 is present at the cell surface, one possibility is that CLDN1 could recruit proapoptotic proteins or complexes and thereby favors apoptosis. Supporting this hypothesis, FasR recruitment by CLDN1 at the membrane is important for enhancing cisplatin-induced apoptosis in epithelial cancer cells KLE and HepG. This work also suggests that CLDN1 could interact with death receptors and induce apoptosis via the extrinsic apoptotic pathway [24]. However, CLDN1 apoptotic protein partners remain to be identified. RhoB could be a candidate to explain the involvement of CLDN1 in apoptosis. Indeed, the Biogrid database indicates that RhoB interacts with ZO-1, ZO-2 and PATJ, which interact themselves with CLDN1 through its PDZ domain binding motif. RhoB is involved in PTX apoptosis induction in fibroblasts [35]. RhoB is also involved in gastric cancer cells response to 5-FU, DOX and cisplatin treatments [36] and in breast cancer cells MCF-7 cisplatin response [37]. Thus, the identification of CLDN1 protein partners in TNBC could provide a better understanding of the involvement of CLDN1 in the chemosensitivity of these tumors.

Our results provided a novel insight about CLDN1 function in TNBC cells chemosensitivity. Furthermore, consistent with this study, a recent report showed that taxanes and anthracyclines neoadjuvant chemotherapy induces CLDN1 expression and decreases Ki-67 of invasive breast tumors [38]. These data highlight that treatments can alter the molecular profile of breast tumor and therefore their chemosensitivity [38]. Several factors are described to regulates CLDN1 expression such as Snail and bHLHE40 which were known as transcriptional repressors of CLDN1 gene [39,40]. These factors were upregulated in breast cancer cells resistant to DOX and their downregulation restore chemosensitivity [41,42]. Moreover, the CLDN1 expression loss could be the results of methylation silencing as it has been established that CLDN1 promoter presented a large number of CpG islands [43,44]. Thus, a better understanding of CLDN1 regulation mechanisms allow the design of new therapeutic strategies in order to induce CLDN1 expression in “claudin-1-low” TNBC and sensitize these very aggressive tumors to clinically commonly used chemotherapeutics agents.

Finally, it has been suggested that CLDN1 could be a predictive marker of chemotherapy response for patients with lung adenocarcinoma [25]. This work supports this hypothesis and suggests that CLDN1 could also constitute a predictive response marker to chemotherapy for TNBC patients.

## 5. Conclusions

This is the first investigation on the effect of CLDN1 expression on TNBC chemosensitivity. Our study demonstrates that CLDN1 is involved in the sensitivity of HCC1806 “claudin-1-high” TNBC cells to 5-FU, PTX and DOX. We also have shown that CLDN1 expression in MDA-MB-231 and Hs578T “claudin-1-low” TNBC cells sensitizes them to these chemotherapeutic agents. In the short term, the therapeutic management of TNBC patients could be improved by using CLDN1 as a predictive marker of response to chemotherapy. Thus, low CLDN1 expression could predict poor response of these tumors to chemotherapeutic agents and could direct towards more appropriates chemotherapeutics treatments by promoting taxanes and anthracycline which induced CLDN1 expression. In the longer term, induction of CLDN1 expression in TNBCs could be proposed as a strategy to increase their sensitivity to chemotherapy.

## Figures and Tables

**Figure 1 cancers-14-05026-f001:**
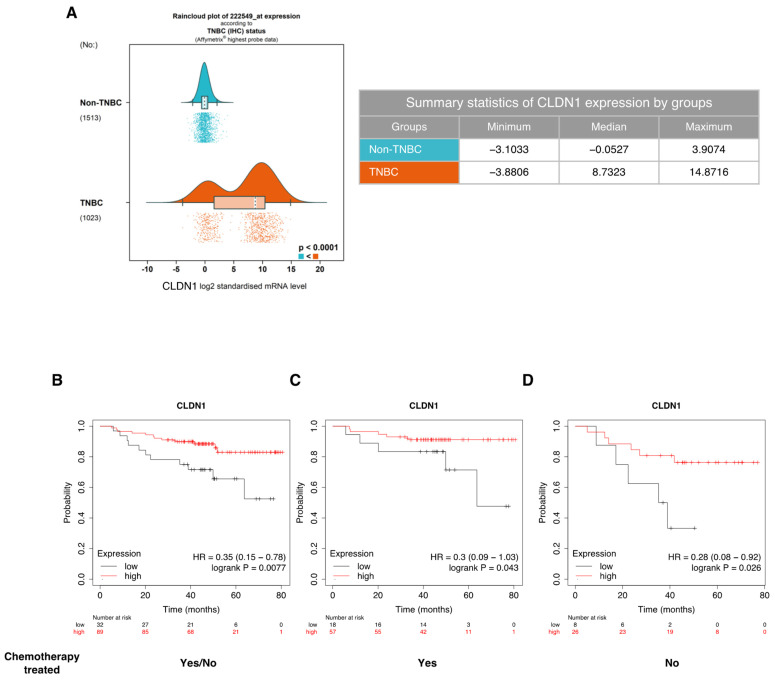
CLDN1 low expression is correlated to a poor prognostic in TNBC patients (**A**) CLDN1 mRNA expression in TNBC versus non-TNBC tumors in Breast Cancer Gene-Expression data (bcGenExMiner v4.8). (**B**–**D**) Kaplan–Meier overall survival curve based on CLDN1 expression and presence and/or absence of chemotherapy treatment. The number of surviving patients is indicated for CLDN1 low and high status at various times (0, 20, 40, 60 and 80 months). Logrank *p*-values < 0.05 for the Kaplan–Meier (KM) plots of CLDN1 gene was considered statistically significant.

**Figure 2 cancers-14-05026-f002:**
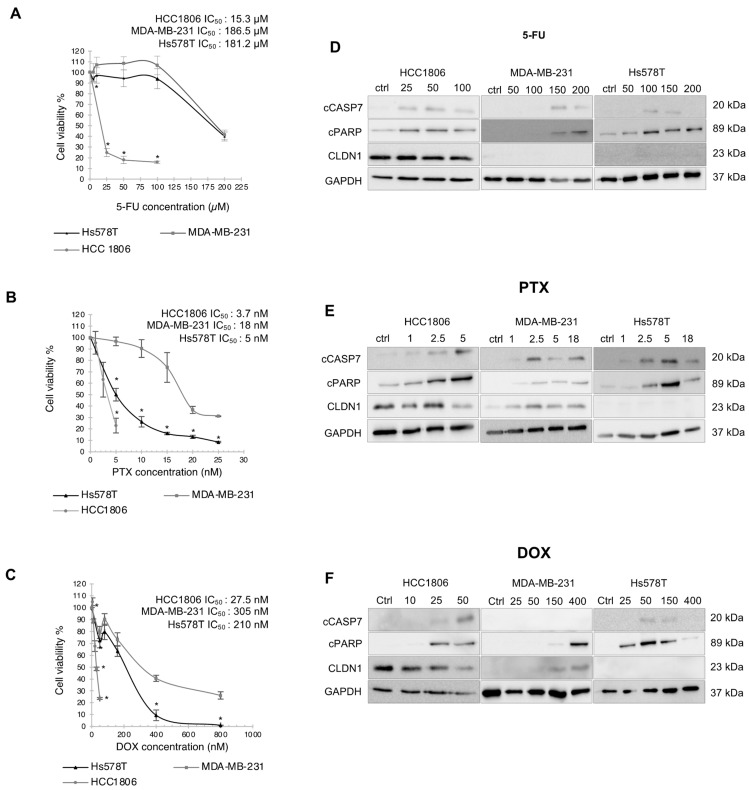
“Claudin-1-high” TNBC cells are more sensitive than “claudin-1-low” TNBC cells to chemotherapy. HCC1806, MDA-MB-231 and Hs578T cells were treated for 72 h with increasing doses of (**A**) 5-FU (0, 5, 10, 25, 50, 100, 200 μM), (**B**) PTX (0, 1, 2.5, 5, 10, 15, 20, 25 nM) and (**C**) DOX (0, 5, 10, 20, 30, 50, 160, 400, 800 nM). Cell viability was determined by measuring ATP level. The results are expressed as a percentage of the control cells exposed to the solvent and IC50s are specified above the graphs. Cells were treated for 72 h at different concentrations of (**D**) 5-FU (0, 25, 50, 100, 150, 200 μM); (**E**) PTX (0, 1, 2.5, 5, 18 nM); and (**F**) DOX (0, 10, 25, 50, 150, 400 nM). Analysis of CLDN1, cleaved apoptotic markers PARP-1 (cPARP) and caspase 7 (cCASP7) expression was performed by western blot. The GAPDH was used as an internal reference. Western Blots expression are representative of 3 to 4 independent experiments. The values represent the means ± SEM of three to four different experiments. A *p*-value of less than 0.05 was considered statistically significant with * *p* < 0.05. The uncropped blots are shown in Appendix A.

**Figure 3 cancers-14-05026-f003:**
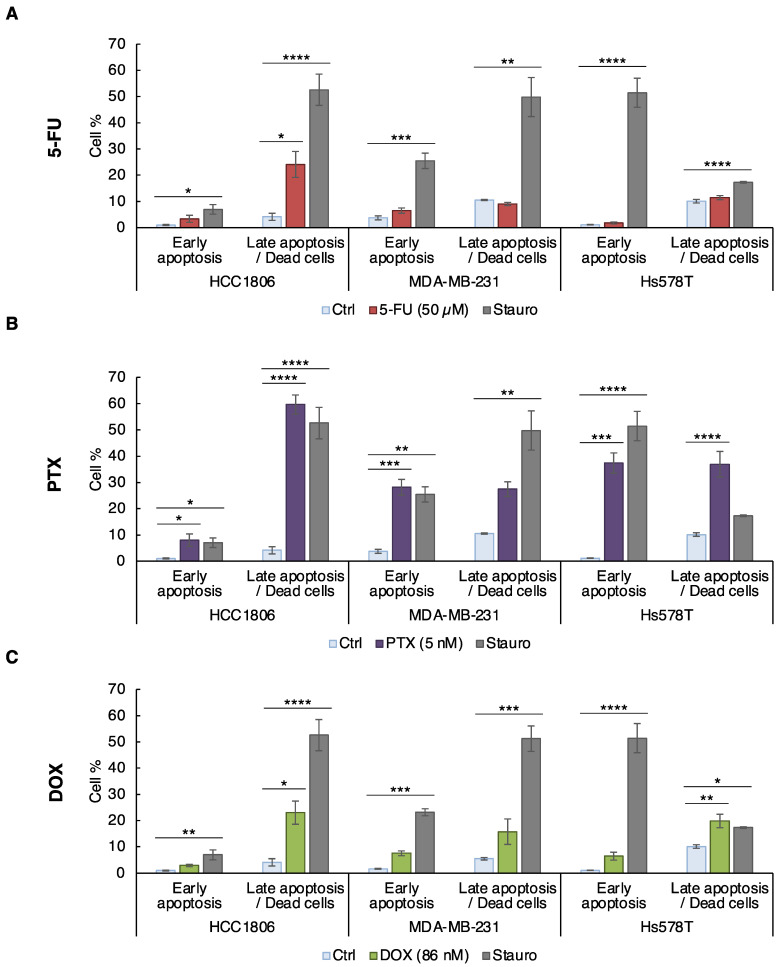
Chemotherapy induced more apoptosis in HCC1806 “claudin-1-high” cells than in “claudin-1-low” cells. HCC1806, MDA-MB-231 and Hs578T cells were treated with (**A**) 50 µM of 5-FU, (**B**) 5 nM of PTX and (**C**) 86 nM of DOX or with solvent (Ctrl) for 72 h. Cells were treated with Staurosporine (Stauro), used as a positive apoptosis induction control, at 50, 500 and 100 nM for each cell line, respectively. The cells are co-labeled with annexin V coupled to FITC and with propidium iodide and the living, early- and late-apoptotic cells were analyzed by flow cytometry. The values represent the means ± SEM of three to four different experiments. A *p*-value of less than 0.05 was considered statistically significant with * *p* < 0.05, ** *p* < 0.01, *** *p* < 0.001, **** *p* < 0.0001.

**Figure 4 cancers-14-05026-f004:**
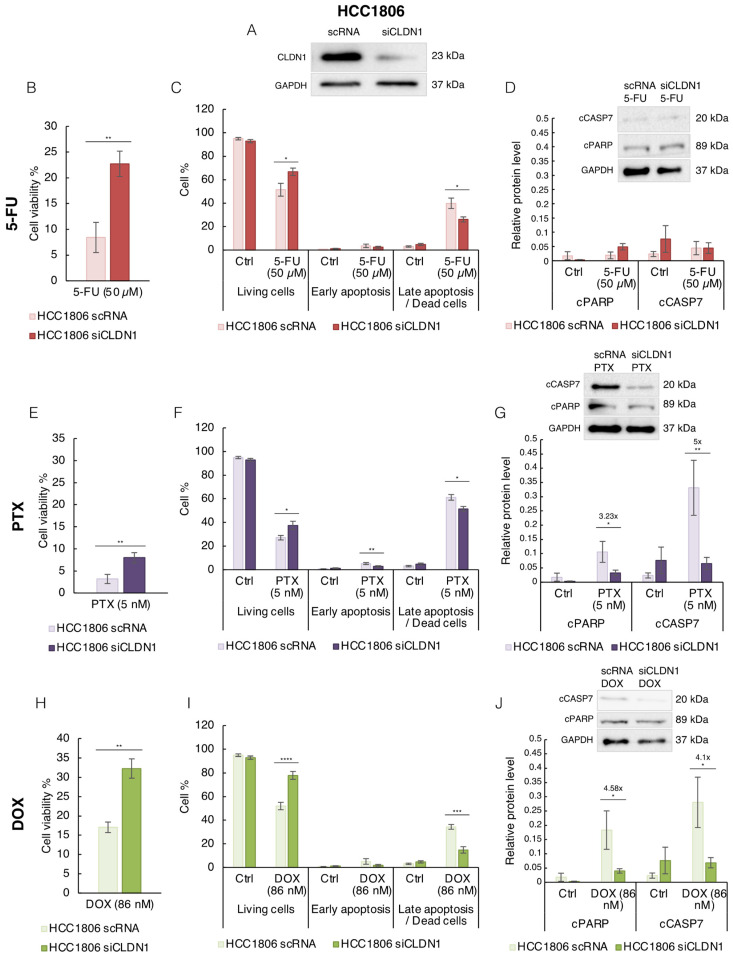
CLDN1 is involved in the chemosensitivity of HCC1806 “claudin-1-high” cells to 5-FU, PTX and DOX. (**A**) HCC1806 cells were transfected with siRNA directed against CLDN1 (siCLDN1) or with a scramble siRNA (scRNA) and CLDN1 expression was studied by western blot. 7 h after transfection, cells were treated for 72 h with (**B**–**D**) 50 μM of 5-FU, (**E**–**G**) 5 nM of PTX and (**H**–**J**) 86 nM of DOX or with solvent (Ctrl). (**B**,**E**,**H**) Cell viability percentage was determined by ATP level measurement and results were expressed as a percentage of the control exposed to the solvent. (**C**,**F**,**I**) HCC1806 cells were co-labeled with annexin V coupled to FITC and with propidium iodide and the living, early- and late-apoptotic cells were analyzed by flow cytometry. (**D**,**G**,**J**) Apoptosis was also analyzed by Western blot by studying the cleavage of PARP-1 (cPARP) and caspase 7 (cCASP7) markers. Relative protein level expressions correspond to the protein of interest on GAPDH ratios. The values represent the means ± SEM of four to eight different experiments. A *p*-value of less than 0.05 was considered statistically significant with * *p* < 0.05, ** *p* < 0.01, *** *p* < 0.001, **** *p* < 0.0001. The uncropped blots are shown in Appendix A.

**Figure 5 cancers-14-05026-f005:**
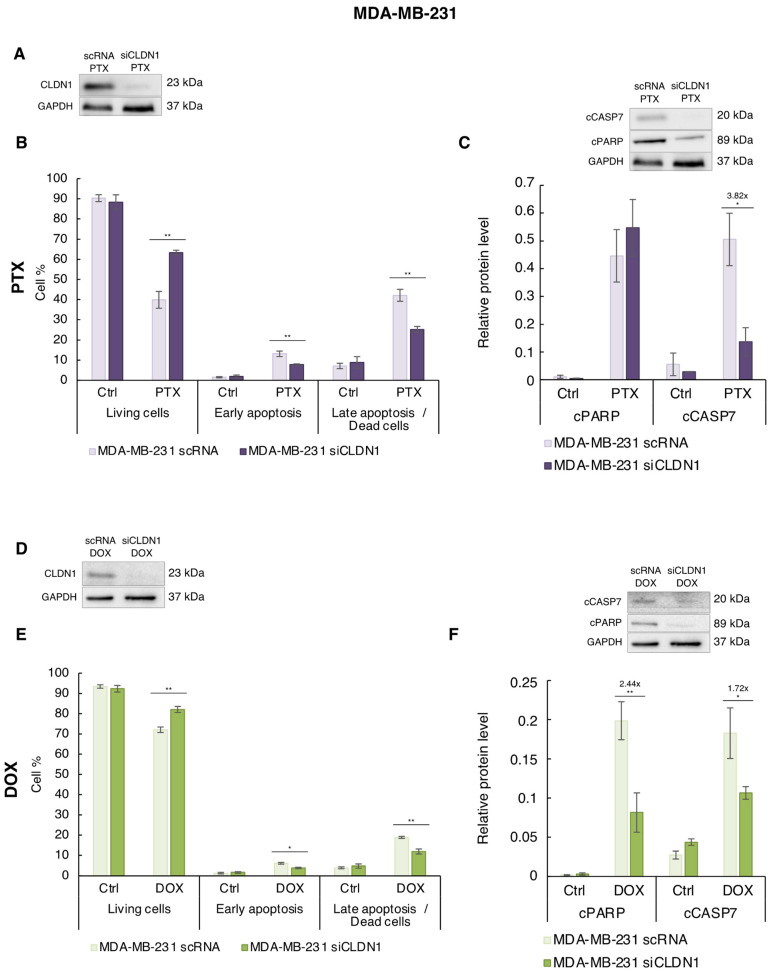
CLDN1 expression is involved in MDA-MB-231 cells sensitivity to PTX and DOX. MDA-MB-231 cells were transfected with a siRNA directed against claudin-1 (siCLDN1) or with a scramble scRNA. 7 h after transfection, cells were treated for 72 h with (**A**–**C**) 5 nM of PTX, (**D**–**F**) 86 nM of DOX or with solvent (Ctrl). (**A**,**D**) CLDN1 expression was studied by western blot. (**B**,**E**) MDA-MB-231 cells were co-labeled with annexin V coupled to FITC and with propidium iodide and the living, early- and late-apoptotic cells were analyzed by flow cytometry. (**C**,**F**) Cleavage of apoptotic markers PARP-1(cPARP) and caspase 7 (cCASP7) expressions were analyzed by Western blot. Relative protein level expressions correspond to the protein of interest on GAPDH ratios. The values represent the means ± SEM of three to five different experiments. A *p*-value of less than 0.05 was considered statistically significant with * *p* < 0.05, ** *p* < 0.01. The uncropped blots are shown in Appendix A.

**Figure 6 cancers-14-05026-f006:**
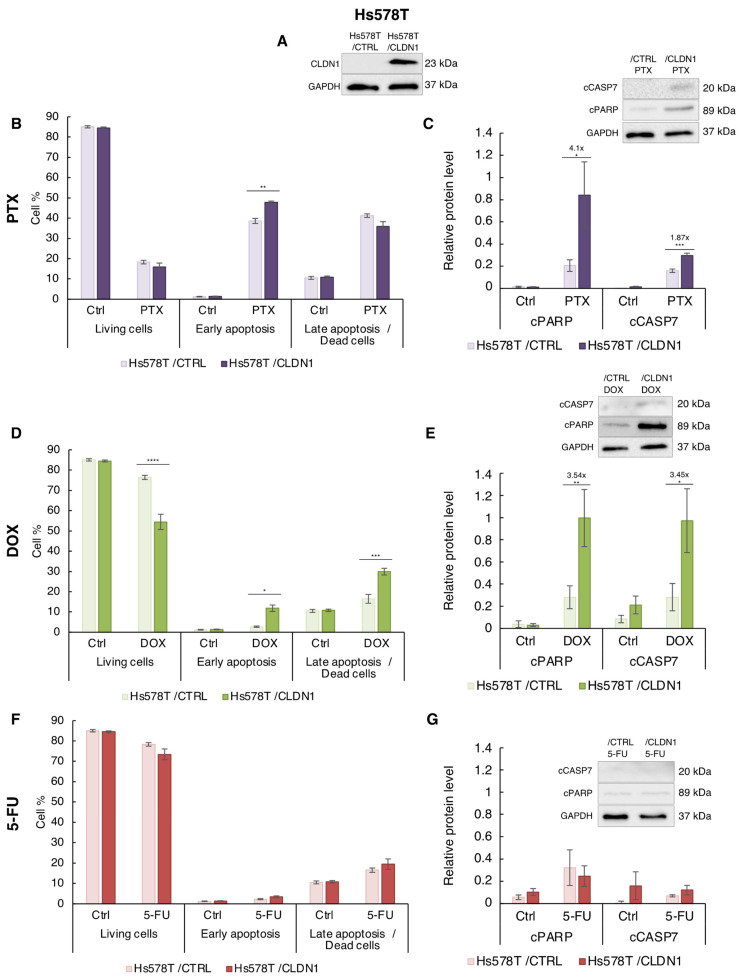
CLDN1 overexpression sensitizes Hs578T cells to PTX and DOX but not to 5-FU. (**A**) Hs578T cells stably transfected with CLDN1 vector (Hs578T/CLDN1) or the empty vector (Hs578T/CTRL) were treated for 72 h with (**B**,**C**) PTX at 5 nM, (**D**,**E**) DOX at 210 nM and (**F**,**G**) 5-FU at 181 µM, or with solvent (Ctrl). (**A**) CLDN1 expression was studied by western blot. (**B**,**D**,**F**) Hs578T cells were co-labeled with annexin V coupled to FITC and with propidium iodide and the living, early- and late-apoptotic cells were analyzed by flow cytometry. (**C**,**E**,**G**) Cleavage of apoptotic markers PARP-1 (cPARP) and caspase 7 (cCASP7) expression were analyzed by Western blot. Relative protein level expressions correspond to the protein of interest on GAPDH ratios. The values represent the means ± SEM of three to four different experiments. A *p*-value of less than 0.05 was considered statistically significant with * *p* < 0.05, ** *p* < 0.01, *** *p* < 0.001, **** *p* < 0.0001. The uncropped blots are shown in Appendix A.

**Figure 7 cancers-14-05026-f007:**
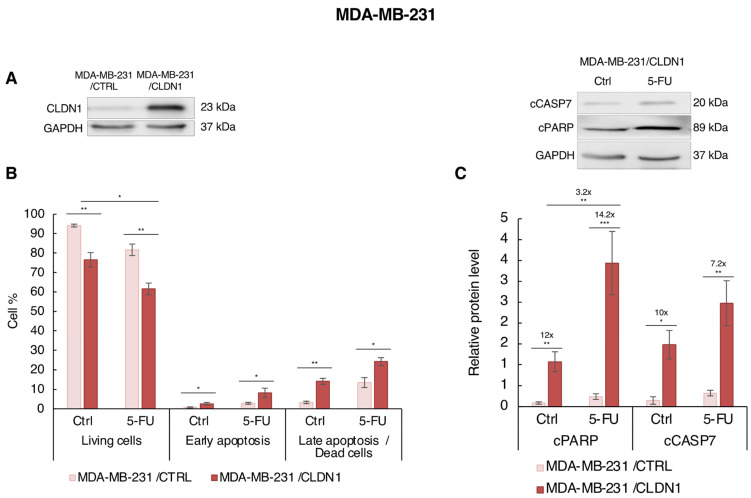
CLDN1 expression sensitizes MDA-MB-231 cells to 5-FU. MDA-MB-231 cells stably transfected with CLDN1 vector (MDA-MB-231/CLDN1) or the empty vector (MDA-MB-231/CTRL) were treated for 72 h with 5-FU at 50 µM or with solvent (Ctrl). (**A**) CLDN1 expression was studied by western blot. (**B**) Cells were co-labeled with annexin V coupled to FITC and with propidium iodide and the living, early- and late-apoptotic cells were analyzed by flow cytometry. (**C**) Cleavage of apoptotic markers PARP-1 (cPARP) and caspase 7 (cCASP7) expressions were analyzed by Western blot. Relative protein level expressions correspond to the protein of interest on GAPDH ratios. The Western blot picture compared MDA-MB-231/CLDN1 non-treated cells (Ctrl) and 5 FU treated cells. The values represent the means ± SEM of three to four different experiments. A *p*-value of less than 0.05 was considered statistically significant with * *p* < 0.05, ** *p* < 0.01, *** *p* < 0.001. The uncropped blots are shown in Appendix A.

## Data Availability

The data presented in this study are available in this article and Appendix A.

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
