# Peer review of "CLDN1 Sensitizes Triple-Negative Breast Cancer Cells to Chemotherapy"

_cancers, 2022, doi:10.3390/cancers14205026_

Round 1

Reviewer 1 Report

Here authors report CLND1 expression sensitizes triple negative breast cancer to anti-cancer drugs. Authors have used 3 different cell lines and 3 different anti-cancer drugs to validate their hypothesis which is appropriate. However, there are minor concerns with this manuscript. 

1: Authors should consider adding clonogenic assays as it would better correlate with drug treatments invivo.

2: Authors should provide flow cytometry cell density charts rather than just quantification bar charts.

3: Has this hypothesis been tested in animal models? Authors should consider validating this hypothesis in animal models.

4: Although authors have showed increase in apoptotic markers cl-casp7 and cl-PARP1 at cell death. Data is too limited; further mechanism investigation into CLND1 expression loss would be beneficial.

5: Finally, line no.# 322   CLDN1 there is a typo mistake. 

Author Response

The reponse is add in a PDF file

Reviewer 2 Report

Overview and general recommendation:

In the manuscript, it is the first time shown that CLDN1 is involved in TNBC chemosensitivity. TNBC “claudin-1-high” cell line (HCC1806) and “claudin-1-low” cell lines (MDA-MB-231 and Hs578T) are used in this assay to test the chemosensitivity with or without chemotherapeutic agents 5-FU, PTX and DOX. The authors also show how overexpression and low-expression of CLDN1 affect the chemosensitivity of these cell lines. This research shows that CLDN1 can increase the sensitivity of TNBC cell lines to 5-FU, PTX and DOX, indicating an important role of CLDN1 in TNBC cell chemosensitivity.

I find the paper is organized in a proper way and the results are well described. The authors perform background research carefully. And major methods are well described in the manuscript and properly used in the research. The figures are well organized and presented in an appropriate way. I suggest the authors include more cell-based data to support their conclusion. And the authors should give more information about how can CLDN1 serve as a chemotherapy response predictive marker for TNBC patients.

Major comments:

1.      In result5 and figure6, the authors claimed that 5-FU induced a strong induction of cPARP and cCASP7 expression in MDA-MB-231/CLDN1 cells. The data of the column figure can support this but the image of western blot in the same figure shows very little change of the expression level of cPARP and cCASP7 upon CLDN1 expression. I think the authors should explain this.

2.      The authors claimed that CLDN1 is involved in apoptosis. I suggest the authors include some cell images indicating the apoptosis status of different cell lines with or without induction of 5-FU, PTX and DOX.

3.      The authors suggest that “CLDN1 could be a chemotherapy response predictive marker for TNBC patients”. The data in the manuscript show that change in CLDN1 expression is not always detectable upon 5-FU, PTX or DOX induction. Can the authors explain more about this?

Minor comments:

1.      Page1 line31, it should be “…and supported the…”.

Author Response

First, we would like to thank the reviewer for the constructing comments that contribute to improve the quality of the manuscript.

Hereafter are our point-by-point answers.

Round 2

Reviewer 1 Report

With current revision, I recommend this manuscript for publication.

Reviewer 2 Report

I find that the authors have put considerable effort into addressing the reports of the referees. They add immunofluorescence images to show the effect of 5-FU, PTX and DOX on cell apoptosis. They also adjust the discussion and conclusion part to clarify that CLDN1 can be a predictive marker of chemotherapy response for patients with lung adenocarcinoma. As a result the paper is very much improved and I have no problem in recommending it for publication.